# RECLAIMING THE SOURCE OF PROGRAMMATIC POLICIES: PROGRAMMATIC VERSUS LATENT SPACES

**Tales H. Carvalho, Kenneth Tjhia, Levi H. S. Lelis**
Amii, Department of Computing Science, University of Alberta
{taleshen,tjhia,levi.lelis}@ualberta.ca

## ABSTRACT

Recent works have introduced LEAPS and HPRL, systems that learn latent spaces of domain-specific languages, which are used to define programmatic policies for partially observable Markov decision processes (POMDPs). These systems induce a latent space while optimizing losses such as the behavior loss, which aim to achieve locality in program behavior, meaning that vectors close in the latent space should correspond to similarly behaving programs. In this paper, we show that the programmatic space, induced by the domain-specific language and requiring no training, presents values for the behavior loss similar to those observed in latent spaces presented in previous work. Moreover, algorithms searching in the programmatic space significantly outperform those in LEAPS and HPRL. To explain our results, we measured the "friendliness" of the two spaces to local search algorithms. We discovered that algorithms are more likely to stop at local maxima when searching in the latent space than when searching in the programmatic space. This implies that the optimization topology of the programmatic space, induced by the reward function in conjunction with the neighborhood function, is more conducive to search than that of the latent space. This result provides an explanation for the superior performance in the programmatic space.

## 1 INTRODUCTION

Programmatic representations of policies for solving reinforcement learning problems can offer important advantages over alternatives, such as neural representations. Previous work showed that due to the inductive bias of the language in which such policies are written, they tend to generalize better to unseen scenarios (Inala et al., 2020; Trivedi et al., 2021). The programmatic nature of policies also allows modularization and reuse of parts of programs (Ellis et al., 2023; Aleixo & Lelis, 2023), which can speed up learning. Previous work also showed that programmatic policies can be more amenable to verification (Bastani et al., 2018) and interpretability (Verma et al., 2018; 2019).

The main challenge with programmatic representations is that, in the synthesis process, one needs to search in very large and often discontinuous policy spaces. While some domain-specific languages (DSLs) are differentiable and gradient descent methods can be used (Qiu & Zhu, 2022; Orfanos & Lelis, 2023), more expressive languages that allow the synthesis of policies with internal states (Inala et al., 2020; Trivedi et al., 2021; Liu et al., 2023) are often full of discontinuities, and thus one must use combinatorial search algorithms to find suitable programs. In an attempt to ease the process of searching for policies, recent work introduced Learning Embeddings for Latent Program Synthesis (LEAPS) (Trivedi et al., 2021), a system that learns a latent space of a DSL with locality in program behavior. That is, if two vectors are near each other in the latent space, then they should decode to programs with similar behavior. Once the latent space is learned, LEAPS uses a local search algorithm to find a latent vector that is decoded into a program encoding a policy for a target task. Liu et al. (2023) extended LEAPS to propose a hierarchical framework, HPRL, to allow the synthesis of programs outside the distribution of programs used to learn the latent space.

In this paper, we evaluate local search algorithms operating in the programmatic space induced by the DSL, and compare them with LEAPS and HPRL. Searching in the original programmatic space involves defining an initial candidate solution (i.e., a program) and a neighborhood function that returns the neighbor programs of a candidate solution. We generate neighbors by following

a process similar to the one used in genetic programming algorithms (Koza, 1992), which was previously used in the context of programmatic policies in multi-agent settings (Medeiros et al., 2022; Aleixo & Lelis, 2023; Moraes et al., 2023). We generate a number of neighbors by modifying parts of the program that represents the candidate. We hypothesized that searching for good policies in the latent space is not easier than in the original programmatic space, as seen in the problems and latent spaces considered in previous work (Trivedi et al., 2021; Liu et al., 2023). Our rationale is that the latent spaces used in previous work are also high-dimensional and non-differentiable, given that the evaluation of latent vectors depends on the execution of the decoded program.

We tested our hypothesis using the same set of problems used to evaluate LEAPS and HPRL. We discovered that a hill-climbing algorithm in the programmatic space, HC, outperformed both LEAPS and HPRL. HC consistently matched or exceeded the performance of the two latent-based methods. To interpret our findings, we examined the value of the behavior loss, used to learn the latent space, within the latent and programmatic spaces, and found that they are similar in each. Although loss values do not account for performance differences between the spaces, they suggest that optimizing solely for the behavior loss does not necessarily produce spaces conducive to search.

We then evaluated the "friendliness" of the two spaces for local search, which is formalized as the probability of a hill-climbing search, which is randomly initialized in the space, converging to a solution with at least a given target reward value. This probability is a measure of the topology of the search space for a given distribution of initial candidates, since it measures the likelihood that the search will be stuck in local maxima. We observed that the programmatic space is never worse and is often much superior to the latent space for a wide range of target reward values. These results not only support our hypothesis that searching in the programmatic space is easier than searching in the latent space, but also suggest that the programmatic space can be more conducive to search.

We conjecture that the effectiveness of latent spaces in synthesizing programmatic policies depends on two properties: how much the latent space compresses the original space and how conducive to search the space is. Intuitively, by compressing the space, the search becomes easier as one has fewer programs to evaluate; by being more conducive to search, the search signal could directly guide the search toward high-return programs. Our empirical results suggest that current systems for learning latent spaces lack either or both of these properties, since the search in the original programmatic space is more effective than the search in latent spaces. The contribution of this paper is to highlight the importance of using a baseline that searches directly in the programmatic space in this line of research. Our baseline allows us to better evaluate and understand the progress in systems that search in latent spaces. The codebase used in this work is available online.[1]

## 1.1 RELATED WORKS

Most of the early work on programmatic policies considered stateless programs, such as decision trees. For example, Verma et al. (2018) and Verma et al. (2019) learn tree-like programs with no internal states. Bastani et al. (2018) use imitation learning to induce decision trees encoding policies. Qiu & Zhu (2022) learn programmatic policies by using a language of differentiable programs, which are identical to oblique decision trees. Learning programmatic policies with internal states, such as programs with while-loops, can be more challenging. This is because the search spaces are often discontinuous and thus not amenable to gradient descent optimization. Inala et al. (2020) presented an algorithm for learning policies in the form of finite-state machines, which can represent loops. Similarly, Trivedi et al. (2021) and Liu et al. (2023) also consider programmatic policies with internal states, which are given by the lines in which the program stops and resumes its execution while interating with the environment. There is also work on programmatic policies in the multi-agent context, where the search spaces are also discontinous and the policies are learned with combinatorial search algorithms (Medeiros et al., 2022; Aleixo & Lelis, 2023; Moraes et al., 2023).

Program synthesis problems also pose problems similar to the ones we discuss (Waldinger & Lee, 1969; Solar-Lezama et al., 2006), where one must search in the programmatic spaces for a program that satisfies the user's intent. These problems can be solved with brute-force search (Udupa et al., 2013; Albarghouthi et al., 2013) or with algorithms guided by a function that is often learned (Odena et al., 2021; Barke et al., 2020; Shi et al., 2022; Ellis et al., 2023; Wong et al., 2021; Ameen & Lelis, 2023). A common method to learning such guiding functions is to use a self-supervised approach

---

[1]https://github.com/lelis-research/prog_policies

where the learning system exploits the structure of the language to generate training data. Similarly to these works, LEAPS can be seen as an attempt to learn a function to help with the search in the programmatic space in the context of reinforcement learning.

## 2 PROBLEM FORMULATION

We consider episodic partially observable Markov decision processes (POMDPs) with deterministic dynamics, deterministic observations, and undiscounted reward functions. This setting can be described by $(\mathcal{S}, \mathcal{A}, \mathcal{O}, p, q, r, S_0)$. In this formulation, $\mathcal{S}$ is the set of states, $\mathcal{A}$ is the set of actions and $\mathcal{O}$ is the set of observations. The function $p : \mathcal{S} \times \mathcal{A} \to \mathcal{S}$ determines the state transition dynamic of the environment, $q : \mathcal{S} \to \mathcal{O}$ the observation given a state and $r : \mathcal{S} \times \mathcal{A} \to \mathbb{R}$ the reward given a state and action. Finally, $S_0$ defines the distribution for the initial state of an episode.

We consider that agents can interact in the environment following policies with internal states. The functions $p$, $q$, and $r$ are hidden from the agent, in the sense that the agent can only observe their output. Policies with internal states are defined by the function $\pi : \mathcal{O} \times \mathcal{H} \to \mathcal{A} \times \mathcal{H}$, where $\mathcal{H}$ represents the internal state of the policy, initialized as a constant $h_0$. Given an initial state $s_0 \sim S_0$ and following the deterministic state transition $s_{t+1} = p(s_t, a_t)$ and the policy $(a_t, h_{t+1}) = \pi(q(s_t), h_t)$ to determine the next states, we can define the trajectory as a function of the policy and initial state $\tau(\pi, s_0) = (a_0, a_1, \ldots, a_T)$ for an episode with $T$ time steps.

The goal of an agent acting in a POMDP is to maximize the cumulative reward over an episode. As the rewards during the episode depend uniquely on the initial state $s_0$ and the policy $\pi$, we can define the return of an episode as $g(s_0, \pi) = \sum_{t=0}^{T} r(s_t, a_t)$. Our objective is to find an optimal policy $\pi^*$ given a policy class $\Pi$.

$$\pi^* = \arg\max_{\pi \in \Pi} \mathbb{E}_{s_0 \sim S_0}[g(s_0, \pi)] \tag{1}$$

### 2.1 PROGRAMMATIC POLICIES

A programmatic policy class $\Pi_{\text{DSL}}$ defines the set of policies that can be represented by a program $\rho$ within a DSL. Figure 1 shows a context-free grammar that defines the DSL for KAREL THE ROBOT, the problem domain we use in our experiments. A context-free grammar is represented by the tuple $(\Sigma, V, R, I)$. Here, $\Sigma$ and $V$ are sets of terminal and non-terminal symbols of the grammar. $R$ defines the set of production rules that can be used to transform a non-terminal symbol into a sequence of terminal and non-terminal ones. Finally, $I$ is the initial symbol of $\mathcal{G}$. In Figure 1, the non-terminal symbols are $\rho, s, b, n, h$ and $a$, where $\rho$ is the initial symbol; terminal symbols include `WHILE`, `REPEAT`, `IF`, etc. An example of a production rule is $a := $ `move`, where the non-terminal $a$ is replaced by the action `move`. This grammar accepts strings defining functions with loops, if-statements, and Boolean functions, such as `frontIsClear`, over the observation space $\mathcal{O}$. The DSL also includes instructions defining actions in the action space $\mathcal{A}$, such as `move` and `turnLeft`. The policy class $\Pi_{\text{DSL}}$ is defined as the set of all programs that the grammar accepts. The problem is to search in the space of programmatic policies $\Pi_{\text{DSL}}$ for a policy that solves Equation 1.

Programs are represented as abstract syntax trees (ASTs). In an AST, each internal node represents a non-terminal symbol and each leaf node a terminal symbol of the grammar. Moreover, each node and its children represent a production rule. For example, for the AST shown in Figure 2, the root and its children represent the production rule $\rho := $ `DEF run m(` $s$ `m)`. The non-terminal $s$ is transformed with the production rule $s := $ `IF c(` $b$ `c) i(` $s$ `i)`. The Boolean expression of the if statement is given by $b := h$ and $h := $ `markersPresent`. Finally, the body of the if statement is given by $s := s; s$, which branches into $s := $ `pickMarker` and $s := $ `move`.

A programmatic policy $\rho \in \Pi_{\text{DSL}}$ is a policy with internal state, where $h_t$ can be interpreted as the pointer in the program $\rho$ after the action taken at time step $t-1$. The internal state $h_t$ and the current observation $q(s_t)$ are sufficient to uniquely determine the action $a_t$ the programmatic policy returns, thus the trajectory a policy generates given an initial state is also deterministic.

```
  Program ρ := DEF run m( s m)
 Statement s := WHILE c( b c) w( s w) | IF c( b c) i( s i) |
                IFELSE c( b c) i( s i) ELSE e( s e) | REPEAT R=n r( s r) |
                s; s | a
 Condition b := h | not( h )
   Number n := 0..19
Perception h := frontIsClear | leftIsClear | rightIsClear |
                markersPresent | noMarkersPresent
   Action a := move | turnLeft | turnRight | putMarker | pickMarker
```

Figure 1: DSL for KAREL THE ROBOT as a context-free grammar.

```
DEF run m(
   IF c( markersPresent c) i(
      pickMarker move
   i)
m)
```

Figure 2: An example of a program defined in KAREL DSL (left) and its AST representation (right). In the AST, MP stands for markersPresent, PM for pickMarker, and M for move.

## 3  SEARCH SPACES FOR PROGRAMMATIC POLICIES

We formalize searching for a programmatic policy as a local search problem. This involves specifying a feasible set $\mathcal{R} \subseteq \Pi_{\text{DSL}}$ and a corresponding neighborhood function $N_K : \mathcal{R} \to \mathcal{R}^K$ which, given a feasible solution $\rho \in \mathcal{R}$, defines its $K$-neighborhood. In this work, we evaluate two search spaces: PROGRAMMATIC SPACE, which uses the DSL directly, and LATENT SPACE, which uses a learned embedding of the DSL.

### 3.1  PROGRAMMATIC SPACE

In this formulation, $\mathcal{R}^{\text{prog}}$ is a subset of the programs the DSL accepts. We define the subset $\mathcal{R}^{\text{prog}}$ by imposing constraints on the size of the programs. In particular, we limit the number of times the production rule $s := s; s$ (statement chaining) can be used, and we also limit the height of the abstract syntax tree (AST) of every program. These constraints are the ones used to determine the distribution of programs used to train the latent space of LEAPS, which we describe in Appendix A.

The $K$-neighborhood of a program $\rho \in \mathcal{R}^{\text{prog}}$, $N_K^{\text{prog}}(\rho)$, consists of $K$ samples of a *mutation* applied in $\rho$. A mutation is defined by uniformly sampling a node $n$ in the AST of $\rho$ and removing one of $n$'s children that represents a non-terminal symbol; the child that is removed is also chosen uniformily at random. In this newly created "hole", we generate a new sub-tree by sequentially sampling a suitable production rule following the probability distribution used to generate the programs to train the latent space of LEAPS (Appendix B). We continue to sample production rules from the grammar until the newly created sub-tree does not have any leaf node representing a non-terminal symbol and the resulting neighbor program is in $\mathcal{R}^{\text{prog}}$. We ignore programs that are not in $\mathcal{R}^{\text{prog}}$ through a sample rejection scheme. That is, if the neighbor of $\rho$ is not in $\mathcal{R}^{\text{prog}}$, we sample a new neighbor until we obtain one that is in $\mathcal{R}^{\text{prog}}$.

### 3.2  LATENT SPACE

LEAPS (Trivedi et al., 2021) and HPRL (Liu et al., 2023) introduce a method for defining a continuous search space for programmatic policies. This is defined by a variational auto-encoder (VAE),

with encoder $Q : \Pi_{\text{DSL}} \to \mathbb{R}^d$ and decoder $P : \mathbb{R}^d \to \Pi_{\text{DSL}}$, trained to reconstruct the text representation of a program $\rho \in \Pi_{\text{DSL}}$. In addition to minimizing a reconstruction loss, LEAPS additionally minimizes two losses related to program behavior, ensuring that programs that yield similar trajectories over a set of initial states are embedded close together in the LATENT SPACE.

As the LATENT SPACE is constructed in a supervised manner, it is trained on a set of programs sampled from a predetermined distribution. For the VAE reconstruction loss, LEAPS uses the same constraints as described for the PROGRAMMATIC SPACE and the same probabilistic DSL rules to generate a training set of programs. For training the VAE behavior losses, LEAPS generates trajectories by running each program from the training set on a set of initial states from the environment.

Given a trained LATENT SPACE, its feasible set $\mathcal{R}^{\text{lat}} \subseteq \Pi_{\text{DSL}}$ is the set of all programs that $P$ can generate. Meanwhile, given a program $\rho = P(z) \in \mathcal{R}^{\text{lat}}$, $z \in \mathbb{R}^d$, each program in its $K$-neighborhood $N_K^{\text{lat},\sigma}(\rho)$ is given by decoding $z + \epsilon$, where $\epsilon \sim \mathcal{N}(0, \sigma I_d)$ and $I_d$ represents the $d \times d$ identity matrix. The standard deviation of the noise $\sigma$ and the dimension $d$ are hyperparameters of the LATENT SPACE.

## 4    LOCAL SEARCH ALGORITHMS

Once the LATENT SPACE is learned, LEAPS relies on the Cross-Entropy Method (CEM) (Rubinstein, 1999) to search for a vector that will decode into a program that approximates a solution to Equation 1. In addition to CEM, we also consider Cross-Entropy Beam Search (CEBS), a method inspired by CEM that retains information about the best candidate solutions from a population. We also consider Hill Climbing (HC), as it is an algorithm that does not offer any mechanism for escaping local minima, and thus can be used to measure properties related to the space topology. In all search algorithms, we break ties arbitrarily.

**Hill Climbing (HC)**   HC starts by sampling a candidate solution, using the probabilistic context-free grammar from Appendix B when searching in the PROGRAMMATIC SPACE, or a vector from distribution $\mathcal{N}(0, I_d)$ when searching in the LATENT SPACE. HC evaluates the $K$-neighborhood set of this initial candidate. If the neighborhood contains another candidate that yields a greater episodic return on the evaluated task than the initial candidate, then this process is repeated from that neighbor. Otherwise, the algorithm returns the best-seen candidate and its episodic return.

**Cross-Entropy Method (CEM)**   CEM generates the $K$ neighbors of an initial candidate, whose latent is sampled from $\mathcal{N}(0, I_d)$, and evaluates all of them in terms of episodic return. CEM then calculates the mean of the latent vectors of the candidate solutions that return the best $E$ episodic returns, where $E$ is a hyperparameter defining the elite size. This process is then repeated from the resulting mean vector. CEM returns the best-found solution once a computational budget is reached.

**Cross-Entropy Beam Search (CEBS)**   CEBS maintains a set of promising candidates, called a beam. Starting from an initial candidate, CEBS generates the $K$ neighbors and selects the best $E$ candidates with respect to their episodic return as the beam of the search. Then CEBS forms the next beam by selecting the top $E$ candidates from the pool given by all $K$ neighbors of the candidates in the beam. This process continues until the mean of the episodic returns seen in the beam does not increase from one iteration to the next. Appendix C presents the pseudocode of HC and CEBS.

## 5    EXPERIMENTS

In this section, we describe our methodology for comparing the PROGRAMMATIC SPACE and the LATENT SPACE with respect to how conducive they are to local search algorithms. We have two sets of experiments. In the first set, we compare CEM searching in the LATENT SPACE, as presented in LEAPS' original paper, CEBS also searching in the LATENT SPACE, HPRL, which implements a hierarchical method over the LATENT SPACE, and HC searching in the PROGRAMMATIC SPACE. In the second set, we compare the spaces in a controlled experiment, where we fix the search algorithm to HC for both spaces. We also explain KAREL THE ROBOT, the domain used in our experiments.

## 5.1 KAREL THE ROBOT DOMAIN

KAREL THE ROBOT was first introduced as a programming learning environment (Pattis, 1994) and, due to its simplified structure, it has recently been adopted as a test-bed for program synthesis and reinforcement learning (Bunel et al., 2018; Chen et al., 2018; Shin et al., 2018; Trivedi et al., 2021). KAREL is a grid environment with local Boolean perceptions and discrete navigation actions.

To define the programmatic policy class for KAREL, we adopt the DSL of Bunel et al. (2018) (Figure 1). This DSL represents a subset of the original KAREL language. Namely, it does not allow the creation of subroutines or variable assignments. The language allows the agent to observe the presence of walls in the immediate neighborhood of the robot, with the perceptions {front|left|right}IsClear, and the presence of markers in the current robot location with markersPresent and noMarkersPresent. The agent can then move the robot with the actions move and turn{Left|Right}, and interact with the markers with {put|pick}Marker.

We consider the KAREL and KAREL-HARD problem sets to define tasks. The KAREL set contains the tasks STAIRCLIMBER, MAZE, FOURCORNERS, TOPOFF, HARVESTER and CLEANHOUSE, all introduced by Trivedi et al. (2021). The KAREL-HARD problem set includes the tasks DOORKEY, ONESTROKE, SEEDER and SNAKE, designed by Liu et al. (2023) as more challenging problems. Trivedi et al. (2021) showed that these domains are challenging for reinforcement learning algorithms using neural representations, so LEAPS and HPRL represent the current state of the art in these problems. A detailed description of each task in both sets is available in Appendix D.

## 5.2 FIRST SET: REWARD-BASED EVALUATION

Our first evaluation reproduces the experiments of Trivedi et al. (2021) and Liu et al. (2023), where we add the results of HC searching in the PROGRAMMATIC SPACE. We use $K = 250$ as the neighborhood parameter for the HC. For CEBS, we set the dimension of the latent vector $d = 256$, the neighborhood size $K = 64$, the elite size $E = 16$, and the noise $\sigma = 0.25$. The hyperparameters for CEM and HPRL are exactly as described in their papers.

For each method, we estimate the expected return of a policy by averaging the returns collected over trajectories starting from a set of initial states. In this experiment, we consider a set of 16 initial states for each problem. We limit the execution of each method to a budget of $10^6$ program evaluations. If an algorithm fails to converge but its execution is still within the budget, we re-sample an initial program and restart the search. We report results over 32 independent runs (seeds) of each method.

Table 1 summarizes our results in the KAREL and KAREL-HARD problem sets, comparing them to the results reported in the papers of LEAPS and HPRL. To better outline the performance of each algorithm, we plot the reached episodic return as a function of the number of episodes in Figure 3[2] and analyze the differences in running time in Appendix E. We further evaluate every method on a harder version of the environments, which were not used in previous work, in Appendix G, and study the impact of the different initialization methods in Appendix H.

We see that HC, based on PROGRAMMATIC SPACE, achieves the highest episodic return on every task compared to all methods based on LATENT SPACE. Furthermore, the plots show that, although HC and CEBS achieve the same mean episodic return at the end of the search process for SEEDER, HC generally does so with a smaller number of samples.

We highlight the results observed on DOORKEY. This is a two-stage task that requires the agent to pick up a marker in a room, yielding a $0.5$ reward, which opens a second room that contains a goal square, which yields an extra $0.5$ reward upon reaching it. LEAPS, HPRL and CEBS are only able to find programmatic policies that achieve $0.5$ episodic return in this task, suggesting that their search procedures reach a local maximum that does not lead to a general solution. Meanwhile, HC achieves a mean episodic return that is higher than $0.5$, suggesting that, in some cases, it is able to escape such local maxima and find policies that reach the final goal. We conjecture that this is a property of the search space itself, since HC does not employ a mechanism to escape local maxima.

---

[2]The CEM curve in this plot is based on our implementation of the algorithm, thus it diverges slightly from the results reported by the LEAPS authors in a few cases.

| Task | LATENT | | | PROGRAMMATIC |
|---|---|---|---|---|
| | LEAPS | HPRL | CEBS | HC |
| STAIRCLIMBER | $\mathbf{1.00} \pm 0.00$ | $\mathbf{1.00} \pm 0.00$ | $\mathbf{1.00} \pm 0.00$ | $\mathbf{1.00} \pm 0.00$ |
| MAZE | $\mathbf{1.00} \pm 0.00$ | $\mathbf{1.00} \pm 0.00$ | $\mathbf{1.00} \pm 0.00$ | $\mathbf{1.00} \pm 0.00$ |
| TOPOFF | $0.81 \pm 0.07$ | $\mathbf{1.00} \pm 0.00$ | $\mathbf{1.00} \pm 0.00$ | $\mathbf{1.00} \pm 0.00$ |
| FOURCORNERS | $0.45 \pm 0.25$ | $\mathbf{1.00} \pm 0.00$ | $\mathbf{1.00} \pm 0.00$ | $\mathbf{1.00} \pm 0.00$ |
| HARVESTER | $0.70 \pm 0.02$ | $\mathbf{1.00} \pm 0.00$ | $\mathbf{1.00} \pm 0.00$ | $\mathbf{1.00} \pm 0.00$ |
| CLEANHOUSE | $0.32 \pm 0.01$ | $\mathbf{1.00} \pm 0.00$ | $\mathbf{1.00} \pm 0.00$ | $\mathbf{1.00} \pm 0.00$ |
| DOORKEY | $0.50 \pm 0.00$ | $0.50 \pm 0.00$ | $0.50 \pm 0.00$ | $\mathbf{0.84} \pm 0.02$ |
| ONESTROKE | $0.65 \pm 0.14$ | $0.80 \pm 0.02$ | $0.90 \pm 0.00$ | $\mathbf{0.95} \pm 0.00$ |
| SEEDER | $0.51 \pm 0.03$ | $0.58 \pm 0.06$ | $\mathbf{1.00} \pm 0.00$ | $\mathbf{1.00} \pm 0.00$ |
| SNAKE | $0.23 \pm 0.06$ | $0.33 \pm 0.07$ | $0.26 \pm 0.01$ | $\mathbf{0.65} \pm 0.03$ |

Table 1: Mean and standard error of final episodic returns of our proposed methods in KAREL and KAREL-HARD problem sets within a budget of $10^6$ program evaluations, compared to the reported results from baselines. LEAPS, HPRL, and CEBS search in LATENT SPACES, while HC searches in PROGRAMMATIC SPACES. HC and CEBS results are estimated over 32 seeds.

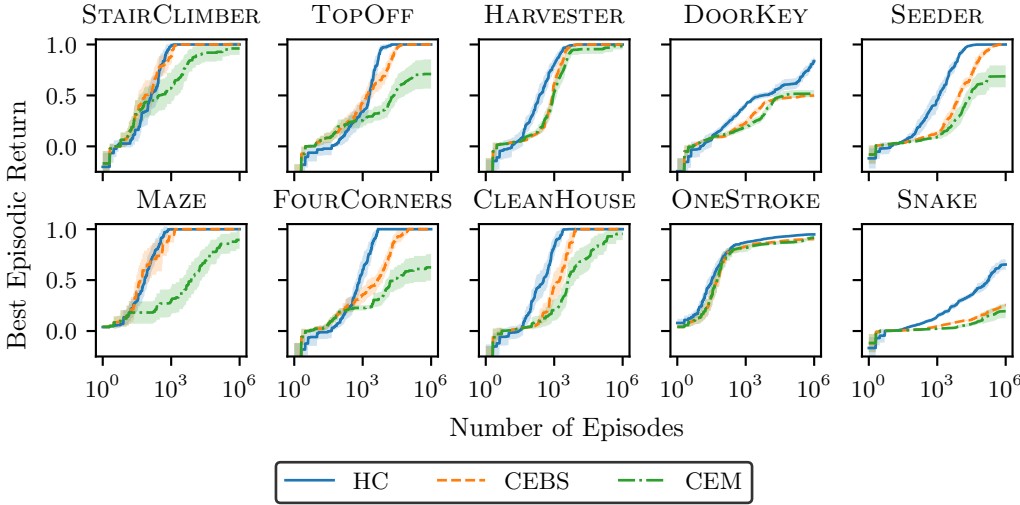

Figure 3: Episodic return performance of all methods in KAREL and KAREL-HARD problem sets. Reported mean and $95\%$ confidence interval over 32 seeds. The x-axis is represented in log scale.

## 5.3 SECOND SET: TOPOLOGY-BASED EVALUATION

To understand the discrepancy between HC and the algorithms searching in the LATENT SPACES, we analyze the PROGRAMMATIC and LATENT SPACE while controlling for the search algorithm.

### 5.3.1 LOCAL BEHAVIOR SIMILARITY ANALYSIS

We first analyze the behavior loss used to train the LATENT SPACE in the two search spaces. We define a metric that measures the loss in the neighborhood of randomly sampled programs in a search space. The similarity between two programmatic policies $\rho$ and $\rho'$, with trajectories from an initial state $s_0$ given by $\tau(\rho, s_0) = (a_0, \ldots, a_T)$ and $\tau(\rho', s_0) = (a'_0, \ldots, a'_{T'})$, as

$$\rho\text{-similarity}(\rho, \rho', s_0) = \frac{\max\{0 \leq t \leq l \mid a_{0:t} = a'_{0:t}\}}{L}, \quad (2)$$

where $l = \min\{T, T'\}$ and $L = \max\{T, T'\}$, and $x_{0:t} = (x_0, \ldots, x_t)$. The $\rho$-similarity returns the normalized length of the longest common prefix of the action sequences $\rho$ and $\rho'$ produced from $s_0$.

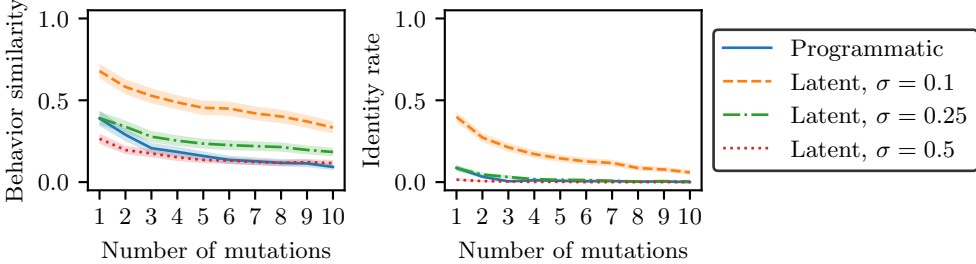

Figure 4: Behavior-similarity and identity-rate metrics on PROGRAMMATIC SPACE and LATENT SPACE ($\sigma = \{0.1, 0.25, 0.5\}$). Reported mean and $95\%$ confidence interval of the estimation of each metric over a set of 32 initial states of the environment and $1,000$ seeds for the initial programs.

The behavior-similarity of a search space defined by the neighborhood function $N_K$ with initial program distribution $P_0$ and initial state distribution $S_0$ is described as

$$\text{behavior-similarity}(N_1, n_{\text{mutations}}) = \mathbb{E}_{\rho_0 \sim P_0, s_0 \sim S_0}[\rho\text{-similarity}(\rho_0, \rho_{n_{\text{mutations}}}, s_0)], \qquad (3)$$

where $\rho_{n_{\text{mutations}}}$ is the outcome of the neighborhood function $N_1$ ($N_K$ with $K = 1$) iteratively applied $n_{\text{mutations}}$ times on $\rho_0$, thus producing a path in the underlying search space.

We observe that measuring behavior-similarity alone can be misleading due to the possibility of observing a neighbor that provides no change to the original program. We propose the metric identity-rate to complement the analysis, which measures the probability of observing a program in its own candidate neighborhood. The identity-rate is defined as follows, where $\mathbb{1}\{\cdot\}$ is the indicator function.

$$\text{identity-rate}(N_1, n_{\text{mutations}}) = \mathbb{E}_{\rho_0 \sim P_0, s_0 \sim S_0}[\mathbb{1}\{\rho_0 = \rho_{n_{\text{mutations}}}\}]. \qquad (4)$$

To estimate the metrics given by Equations 3 and 4, we sample a set of 32 initial states from a distribution $S_0$ and a set of $1,000$ initial programs from $P_0$. $S_0$ is composed of random KAREL maps unrelated to any task in the problem sets, and we set $P_0$ differently for each search space. For PROGRAMMATIC SPACE, $P_0$ is given by the probabilistic DSL rules from Appendix B, and for LATENT SPACE, $P_0$ is given by $\mathcal{N}(0, I_d)$. We run the metrics estimations as a function of $n_{\text{mutations}} \in [1, 10]$ and three specifications of LATENT SPACE, setting $\sigma = \{0.1, 0.25, 0.5\}$—hyperparameters commonly used by LEAPS and HPRL. The results are presented in Figure 4.

Although the behavior-similarity metric is one of the objectives of the LATENT SPACE, the PROGRAMMATIC SPACE achieves comparable behavior-similarity values. Although the setting $\sigma = 0.1$ on LATENT SPACE achieves high behavior-similarity, in practice it is not necessarily more conducive to search, possibly due to its higher identity rate. However, the observed result still does not explain the performance discrepancy that we observe when searching for policies that solve tasks.

### 5.3.2 CONVERGENCE ANALYSIS

Next, we look at the topology of each search space with respect to the return function of the tasks we want to solve. Specifically, we want to measure how conducive a given space is to search. To do so, we use a search algorithm that cannot escape local minima, HC, and measure how likely the search is to converge to a solution of a given quality in the search space of interest.

We define the convergence rate of a search space given by the neighborhood function $N_K$, with initial program distribution $P_0$ and initial state distribution $S_0$ of a given POMDP. The initial program distribution for the PROGRAMMATIC SPACE is given by the LEAPS probabilistic context-free grammar (Appendix B); for the LATENT SPACE, it is given by the programs one decodes after sampling a latent vector from $\mathcal{N}(0, I_d)$. The convergence rate is measured in terms of $g_{\text{target}} \in [0, 1]$ as follows.

$$\text{convergence-rate}(N_K, g_{\text{target}}) = \mathbb{E}_{\rho_0 \sim P_0, s_0 \sim S_0}[\mathbb{1}\{g_{\text{search}}(\rho_0, s_0) \geq g_{\text{target}}\}], \qquad (5)$$

where $g_{\text{search}}(\rho_0, s_0)$ is the return of the best-performing program encountered in the search starting with candidate $\rho_0$ and with the return computed by rolling the policies out from $s_0$.

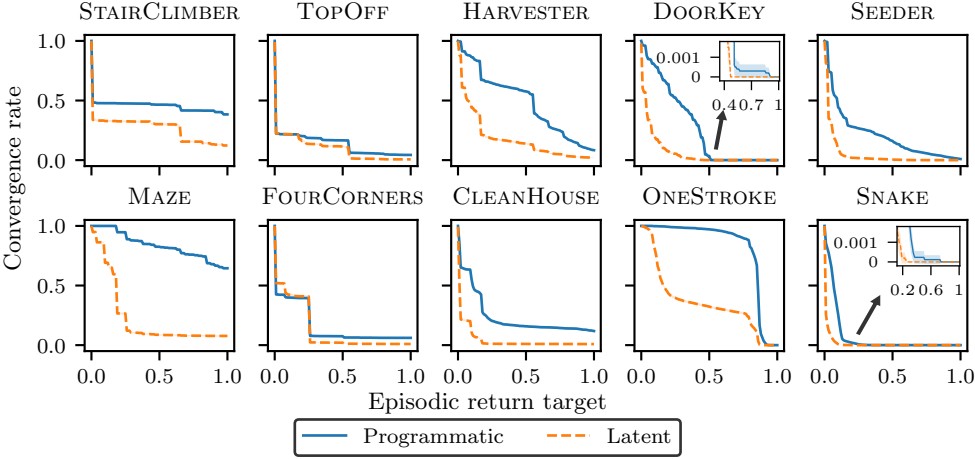

Figure 5: Convergence rate of the hill-climbing algorithm in the PROGRAMMATIC SPACE and in the LATENT SPACE with neighborhood size $K = 250$. Reported mean and $95\%$ confidence interval of estimation over a set of $10,000$ initial candidates. The plots for DOORKEY and SNAKE show a zoomed-in region highlighting runs of the search that achieve reward values larger than $0.5$.

As the return of the program depends on the task we are solving, we report estimates of Equation 5 for each task. The estimation involves sampling a set of 32 states given by the task's initial state distribution, and sampling $10,000$ programs to serve as initial candidates for search. In this experiment, $\sigma$ follows the values used in the original LEAPS experiments for each task: $\sigma = 0.5$ for FOUR-CORNERS and HARVESTER, $\sigma = 0.1$ for MAZE, and $\sigma = 0.25$ for all other tasks. The estimation of convergence-rate of PROGRAMMATIC SPACE and LATENT SPACE, both set to a neighborhood size $K = 250$, for every task in our problem sets is shown in Figure 5 as a function of $g_{\text{target}} \in [0, 1]$, using HC as the search algorithm. In the figure, we show a zoomed-in plot for the tasks DOORKEY and SNAKE to better visualize cases with low convergence rate. Table 1 and Figure 3 show that the search in the programmatic space achieves reward values larger than $0.5$ for DOORKEY and SNAKE; the zoomed-in regions in Figure 5 show that these are rare events, but possible to be observed with a reasonable search budget. We show in Appendix K that policies with a return greater than $0.5$ exist in the LATENT SPACE for DOORKEY; it is the search that failed to find them. The plots show that, even in tasks where HC only matched or performed marginally better than latent methods, the PROGRAMMATIC SPACE is more likely to yield policies with greater episodic return. This suggests that this search space is more conducive to search algorithms than the LATENT SPACE.

## 6 CONCLUSION

In this paper, we showed that, despite recent efforts in learning latent spaces to replace programmatic spaces, the latter can still be more conducive to search. Empirical results in KAREL THE ROBOT showed that a simple hill-climbing algorithm searching in the programmatic space can significantly outperform the current state-of-the-art algorithms that search in latent spaces. We measured both the learned latent space and the programmatic space in terms of the loss function used to train the former. We discovered that both have similar loss values, even though the programmatic space does not require training. We also compared the topology of the two spaces through the probability of a hill-climbing search being stuck at local maxima in the two spaces, and found that the programmatic space is more conducive to search. Our results suggest that the use of the original programmatic space is an important baseline that was missing in previous work. Our results also suggest that learning latent spaces for easing the process of synthesizing programmatic policies for solving reinforcement learning problems is still an open and challenging research question.

ACKNOWLEDGEMENTS

This research was supported by Canada's NSERC and the CIFAR AI Chairs program, and enabled in part by support provided by the Digital Research Alliance of Canada. We thank the reviewers for their helpful discussions and suggestions, and the Area Chair who handled our paper, for following the discussions closely and taking the time to understand the contributions of our work. Finally, we also thank Matthew Guzdial and Adam White for their comments on an early version of this work.

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

$$\Pr(s := \texttt{WHILE}) = 0.15; \Pr(s := \texttt{IF}) = 0.08; \Pr(s := \texttt{IFELSE}) = 0.04;$$
$$\Pr(s := \texttt{REPEAT}) = 0.03; \Pr(s := s; s) = 0.5; \Pr(s := a) = 0.2;$$
$$\Pr(b := h) = 0.9; \Pr(b := \texttt{not ( } h \texttt{ )}) = 0.1;$$
$$\Pr(n := 0) = \Pr(n := 1) = \cdots = \Pr(n := 19) = 1/20;$$
$$\Pr(h := \texttt{frontIsClear}) = 0.5; \Pr(h := \texttt{leftIsClear}) = 0.15;$$
$$\Pr(h := \texttt{rightIsClear}) = 0.15; \Pr(h := \texttt{markersPresent}) = 0.1;$$
$$\Pr(h := \texttt{noMarkersPresent}) = 0.1;$$
$$\Pr(a := \texttt{move}) = 0.5; \Pr(a := \texttt{turnLeft}) = 0.15; \Pr(a := \texttt{turnRight}) = 0.15;$$
$$\Pr(a := \texttt{pickMarker}) = 0.1; \Pr(a := \texttt{putMarker}) = 0.1.$$

Figure 6: Adopted probabilities for the KAREL THE ROBOT DSL as a probabilistic context-free grammar.

## A SETTINGS FOR GENERATING PROGRAMS

In this work, we use the same constraints for generating programs as the LEAPS project (Trivedi et al., 2021), as described below.

- Maximum AST height: 4;
- Maximum statement chaining ($s := s; s$ rule): 6;
- Maximum program length (in number of symbols in the program text representation): 45.

## B PROBABILITIES FOR DSL PRODUCTION RULES

We adopt a fixed probability for each DSL production rule, described in Figure 6 as a probabilistic context-free grammar. The adopted probabilities are based on the LEAPS project specifications (Trivedi et al., 2021).

## C ALGORITHM DETAILS OF SEARCH ALGORITHMS

We present pseudo-code implementations of HC and CEBS as described in Section 4 in Algorithms 1 and 2, respectively.

---

**Algorithm 1** Hill Climbing for Programmatic Policies

---

**Require:** $N$, the neighborhood function; $K$, the neighborhood size; $P_0$, the initial program distribution; $g$: the task return function; $\mathcal{S}_0$, the set of task initial states.
**Ensure:** $\bar{\rho}$, best-seen program with respect to highest episodic return estimate; $\bar{g}$, estimated episodic return of best-seen program.
1: $\bar{\rho} \sim P_0$
2: $\bar{g} \leftarrow \frac{1}{|\mathcal{S}_0|} \sum_{s_0 \in \mathcal{S}_0} g(\pi_{\bar{\rho}}, s_0)$
3: **repeat**
4:     in_local_maximum $\leftarrow$ true
5:     **for** each $\rho$ in $N_K(\bar{\rho})$ **do**
6:         $g \leftarrow \frac{1}{|\mathcal{S}_0|} \sum_{s_0 \in \mathcal{S}_0} g(\pi_{\rho}, s_0)$
7:         **if** $g > \bar{g}$ **then**
8:             $\bar{\rho} \leftarrow \rho$
9:             $\bar{g} \leftarrow g$
10:            in_local_maximum $\leftarrow$ false
11: **until** in_local_maximum = true

---

---

**Algorithm 2** Cross-Entropy Beam Search for Programmatic Policies in LATENT SPACES

---

**Require:** $N^{\text{lat}}$, the neighborhood function; $K$, the neighborhood size; $E$, the size of the beam; $\sigma$, the noise parameter for the LATENT SPACE; $P_0$, the initial program distribution; $g$: the task return function; $\mathcal{S}_0$, the set of task initial states.

**Ensure:** $\bar{\rho}$, best-seen program with respect to highest episodic return estimate; $\bar{g}$, estimated episodic return of best-seen program.

1: $\bar{\rho} \sim P_0$
2: $\bar{g} \leftarrow \frac{1}{|\mathcal{S}_0|} \sum_{s_0 \in \mathcal{S}_0} g(\pi_{\bar{\rho}}, s_0)$
3: best_mean_elite_return $\leftarrow -\infty$
4: candidates $\leftarrow N_K^{\text{lat},\sigma}(\bar{\rho})$
5: **repeat**
6:     in_local_maximum $\leftarrow$ true
7:     $\mathbf{g} \leftarrow []$
8:     **for** each $\rho$ in candidates **do**
9:         $\mathbf{g}$.append($\frac{1}{|\mathcal{S}_0|} \sum_{s_0 \in \mathcal{S}_0} g(\pi_\rho, s_0)$)
10:         **if** $\mathbf{g}[-1] > \bar{g}$ **then**                     ▷ Index -1 represents the last element in the list.
11:             $\bar{\rho} \leftarrow \rho$
12:             $\bar{g} \leftarrow \mathbf{g}[-1]$
13:     elite_indices $\leftarrow$ argtop-$E(\mathbf{g})$
14:     **if** $\frac{1}{E} \sum_{i \in \text{elite\_indices}} \mathbf{g}[i] >$ best_mean_elite_return **then**
15:         best_mean_elite_return $\leftarrow \frac{1}{E} \sum_{i \in \text{elite\_indices}} \mathbf{g}[i]$
16:         in_local_maximum $\leftarrow$ false
17:     candidates $\leftarrow \bigcup_{i \in \text{elite\_indices}} N_{K/E}^{\text{lat},\sigma}(\rho[i])$     ▷ Aggregates as a $K$-neighborhood of the elite.
18: **until** in_local_maximum = true

---

# D KAREL PROBLEM SETS

In this Section, we specify the initial state and return function of every task in KAREL and KAREL-HARD problem sets. Further details of each task are present in LEAPS (Trivedi et al., 2021) and HPRL (Liu et al., 2023), works that introduced KAREL and KAREL-HARD, respectively. All tasks time out an episode after $10,000$ actions.

## D.1 KAREL

**STAIRCLIMBER**   This environment is given by a $12 \times 12$ grid with stairs formed by walls. The agent starts on a random position on the stairs and its goal is to reach a marker that is also randomly initialized on the stairs. If the agent reaches the marker, the agent receives 1 as an episodic return and 0 otherwise. If the agent moves to an invalid position, i.e. outside the contour of the stairs, the episode terminates with a $-1$ return.

**MAZE**   A random maze is initialized on an $8 \times 8$ grid, and a random marker is placed on an empty square as a goal. The agent starts on a random empty square of the grid and its goal is to reach the marker goal, which yields a 1 episodic return. Otherwise, the agent receives 0 as a return.

**TOPOFF**   Markers are placed randomly on the bottom row of an empty $12 \times 12$ grid. The goal of the agent, initialized on the bottom left of the map, is to place one extra marker on top of every marker on the map. The return of the episode is given by the number of markers that have been topped off divided by the total number of markers.

**FOURCORNERS**   Starting on a random cell on the bottom row of an empty $12 \times 12$ grid, the goal of the agent is to place one marker in each corner of the map. Return is given by the number of corners with one marker divided by four.

**HARVESTER**   The agent starts on a random cell on the bottom row of an $8 \times 8$ grid, that starts with a marker on each cell. The goal of the agent is to pick up every marker on the map. Return is given by the number of picked-up markers divided by the total number of markers.

| | PROGRAMMATIC SPACE | LATENT SPACE |
|---|---|---|
| Elapsed time (seconds) | $0.0021 \pm 0.0002$ | $0.0293 \pm 0.0004$ |

Table 2: Time for generating one neighbor from a given candidate, measured over $1,000$ initial random candidates. Reported mean and $95\%$ confidence interval.

**CLEANHOUSE**  In this task, the agent starts on a fixed cell of a complex $14 \times 22$ grid environment made of many connected rooms, with ten markers randomly placed adjacent to the walls. The goal of the agent is to pick up every marker on the map and the return is given by the number of picked-up markers divided by the total number of markers.

### D.2  KAREL-HARD

**DOORKEY**  The agent starts on a random position on the left side of an $8 \times 8$ grid that is vertically split into two chambers. The agent goal is to pick up a marker on the left chamber, which opens a door connecting both chambers and allows the agent to reach a goal marker. Picking up the first marker yields a $0.5$ reward, and reaching the goal yields an additional $0.5$.

**ONESTROKE**  Starting on a random position of an empty $8 \times 8$ grid, the goal of the agent is to visit every grid cell without repeating. Visited cells become a wall that terminates the episode upon touching. The episodic return is given by the number of visited cells divided by the total number of cells in the initial state.

**SEEDER**  The environment starts as an empty $8 \times 8$ grid, with the agent placed randomly in any square. The agent's goal is to place one marker in every empty cell of the map. The return is given by the number of cells with one marker divided by the total number of empty cells at the start of the episode.

**SNAKE**  In this task, the agent and one marker are randomly placed on an empty $8 \times 8$ grid. The agent acts like the head of a snake, whose body grows each time a marker is collected. The goal of the agent is to touch the marker on the map without colliding with the snake's body, which terminates the episode. Each time the marker is collected, it is placed in a new random location, until 20 markers are collected. The episodic return is given by the number of collected markers divided by 20.

## E  RUNNING TIME COMPARISON OF PROGRAMMATIC AND LATENT SPACES

In this section, we compare the neighborhood generation process of each search space in terms of running time. We do this by measuring the time the PROGRAMMATIC SPACE and the LATENT SPACE take to generate one neighbor from a given candidate program, sampled from the initial distribution, and present the results in Table 2. We see that sampling from the programmatic space is more than 10 times faster than sampling from the latent space. This means that the gap between the search in PROGRAMMATIC SPACE and LATENT SPACE would be larger if the results reported in the paper were in terms of running time instead of episodes.

## F  EXAMPLES OF OBTAINED SOLUTIONS

In this section, we show representative examples of programmatic policies from HC and CEBS across some relevant tasks. We selected programs that yield the highest return for each algorithm. Results are presented in Tables 3 and 4 for HC and CEBS, respectively.

## G  EVALUATION ON CRASHABLE KAREL

To further evaluate the search algorithms, we propose a modification of the KAREL environment. In this version, which we name CRASHABLE, invalid actions terminate episodes. This change implies

| Task | Solution | Return |
|------|----------|--------|
| HARVESTER | DEF run m( WHILE c( leftIsClear c) w( WHILE c( leftIsClear c) w( REPEAT R=14 r( move pickMarker r) turnRight w) WHILE c( rightIsClear c) w( pickMarker turnRight move turnLeft w) WHILE c( frontIsClear c) w( move w) w) m) | 1.0 |
| CLEANHOUSE | DEF run m( WHILE c( leftIsClear c) w( move turnRight move move w) WHILE c( frontIsClear c) w( turnRight w) WHILE c( noMarkersPresent c) w( move REPEAT R=7 r( turnLeft move pickMarker r) w) move turnLeft m) | 1.0 |
| DOORKEY | DEF run m( WHILE c( frontIsClear c) w( move w) turnLeft move WHILE c( noMarkersPresent c) w( turnRight move move w) IF c( leftIsClear c) i( pickMarker move move WHILE c( noMarkersPresent c) w( move turnRight move w) putMarker i) m) | 1.0 |
| ONESTROKE | DEF run m( IF c( frontIsClear c) i( turnRight i) WHILE c( noMarkersPresent c) w( WHILE c( frontIsClear c) w( turnRight move w) turnLeft IFELSE c( frontIsClear c) i( move turnRight pickMarker move move move i) ELSE e( turnRight move e) w) m) | 0.953 |
| SEEDER | DEF run m( turnLeft WHILE c( noMarkersPresent c) w( putMarker REPEAT R=10 r( move r) REPEAT R=5 r( WHILE c( markersPresent c) w( turnLeft move turnRight w) pickMarker r) w) WHILE c( frontIsClear c) w( turnLeft w) m) | 1.0 |
| SNAKE | DEF run m( turnLeft WHILE c( frontIsClear c) w( move w) WHILE c( rightIsClear c) w( WHILE c( rightIsClear c) w( move turnLeft move IF c( frontIsClear c) i( move move i) turnLeft w) putMarker WHILE c( rightIsClear c) w( putMarker w) turnRight w) m) | 1.0 |

Table 3: Representative high-return solutions from HC searches in the PROGRAMMATIC SPACE.

| Task | Solution | Return |
|------|----------|--------|
| HARVESTER | `DEF run m( WHILE c( leftIsClear c) w( move pickMarker move turnLeft pickMarker move pickMarker move turnLeft move pickMarker move turnLeft move pickMarker move w) m)` | 1.0 |
| CLEANHOUSE | `DEF run m( WHILE c( noMarkersPresent c) w( move pickMarker turnLeft w) WHILE c( leftIsClear c) w( move move w) WHILE c( frontIsClear c) w( move move w) m)` | 1.0 |
| DOORKEY | `DEF run m( WHILE c( rightIsClear c) w( turnLeft pickMarker w) WHILE c( noMarkersPresent c) w( turnRight move w) pickMarker WHILE c( noMarkersPresent c) w( turnRight move w) putMarker m)` | 0.6875 |
| ONESTROKE | `DEF run m( WHILE c( noMarkersPresent c) w( turnLeft move turnLeft WHILE c( frontIsClear c) w( turnLeft move w) pickMarker move move move w) WHILE c( noMarkersPresent c) w( turnLeft move w) pickMarker move move move move m)` | 0.9288 |
| SEEDER | `DEF run m( WHILE c( noMarkersPresent c) w( putMarker move move WHILE c( markersPresent c) w( turnRight move w) w) m)` | 1.0 |
| SNAKE | `DEF run m( WHILE c( noMarkersPresent c) w( REPEAT R=13 r( IFELSE c( frontIsClear c) i( turnRight move pickMarker i) ELSE e( move pickMarker REPEAT R=13 r( turnRight move r) REPEAT R=13 r( pickMarker r) move e) pickMarker r) w) m)` | 0.4375 |

Table 4: Representative high-return solutions from CEBS search in LATENT SPACE.

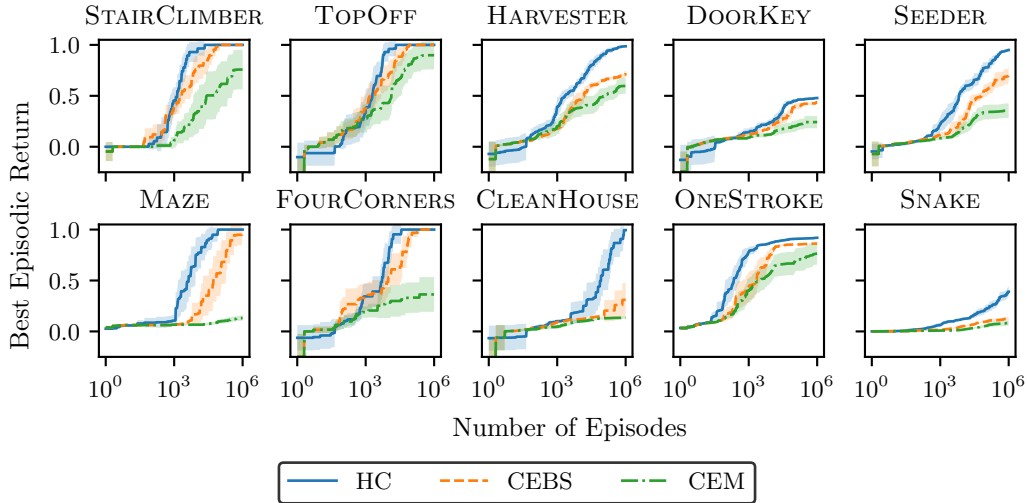

Figure 7: Episodic return performance of all methods in KAREL and KAREL-HARD problem sets, using the CRASHABLE version of the environment, that does not allow invalid actions. Reported mean and 95% confidence interval over 16 executions. The x-axis is represented in a log scale for better visualization.

that the same tasks from KAREL and KAREL-HARD become more difficult to solve, as the valid trajectories are stricter. Figure 7 outlines the episodic return performance of CEM, CEBS, and HC on the CRASHABLE version of every task of the problem sets.

The results show a greater discrepancy between the programmatic and latent methods. We conjecture that this is because the LATENT SPACE was trained with trajectories obtained from the original environment setting and it is unable to generalize to the CRASHABLE environment. Since the PROGRAMMATIC SPACE does not require any training, it generalizes better to the CRASHABLE setting.

## H    EVALUATING THE IMPACT OF INITIALIZATION METHODS

To measure the impact of the initialization methods of the search algorithms, we evaluate an alternative version of all search algorithms using the initialization rule of the opposed search space. For CEM and CEBS, this version of the algorithms initializes the search with policies sampled from the defined probabilistic context-free grammar. And for HC in PROGRAMMATIC SPACE, search is initialized by decoding a latent sampled from $\mathcal{N}(0, I_d)$ in the LATENT SPACE. Figure 8 compares the performance of HC in PROGRAMMATIC SPACE with the alternative initialization scheme, while Figures 9 and 10 compare the performance of CEM and CEBS, respectively, with the alternative initialization scheme.

Results show that the performance of the alternative version of the algorithms was very similar to the original algorithm in most cases, and marginally inferior in others. This suggests that both initialization methods are similar and one does not provide a significant advantage over the other.

## I    CONVERGENCE RATE FOR DIFFERENT NEIGHBORHOOD SIZES

We expand the convergence rate analysis by adopting neighborhood functions $N_K$ with different neighborhood sizes $K$ in Equation (5). The convergence-rate analysis on a space given by a lower $K$ is related to how robust the search space is to conduct a search process, given that it relies on a small number of samples. On the other hand, the result of convergence-rate on higher $K$ gives us information about how capable the search space is, as it can expand a search state further to find the better candidate. Figure 11 compares the convergence-rate estimation of both PROGRAMMATIC and LATENT SPACE adopting $K = \{10, 250, 1000\}$, and suggests that the PROGRAMMATIC SPACE is

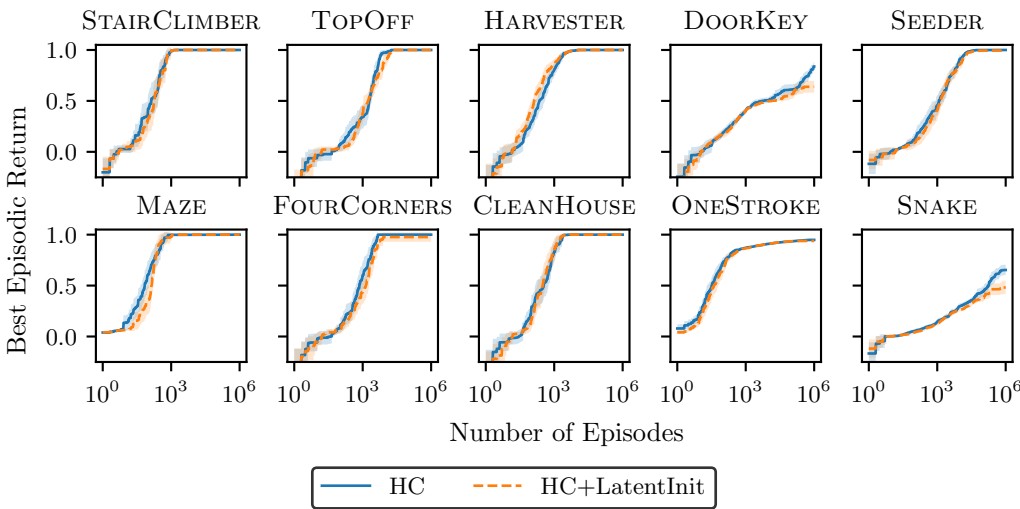

Figure 8: Episodic return performance of HC in PROGRAMMATIC SPACE with original and latent initialization in KAREL and KAREL-HARD problem sets. Reported mean and 95% confidence interval over 32 seeds. The x-axis is represented in a log scale for better visualization.

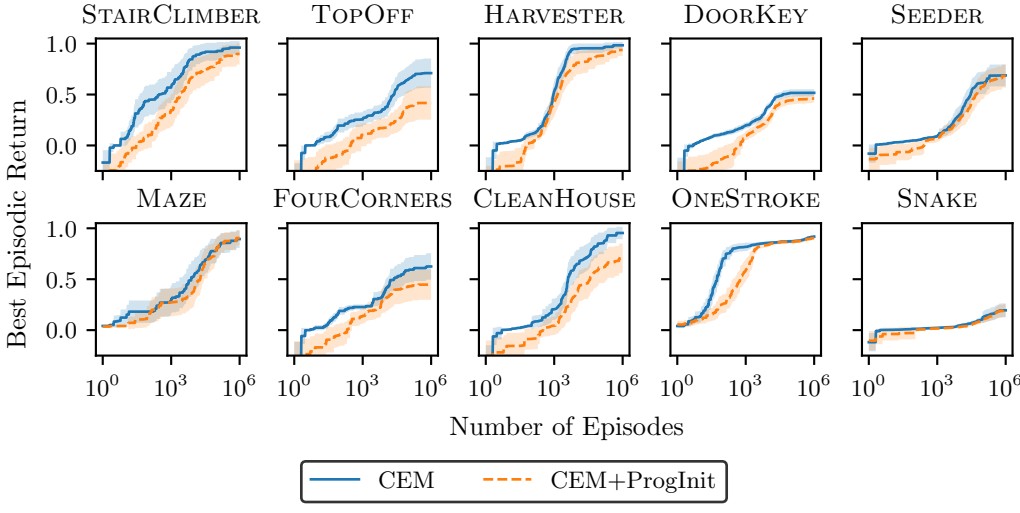

Figure 9: Episodic return performance of CEM in LATENT SPACE with original and programmatic initialization in KAREL and KAREL-HARD problem sets. Reported mean and 95% confidence interval over 32 seeds. The x-axis is represented in a log scale for better visualization.

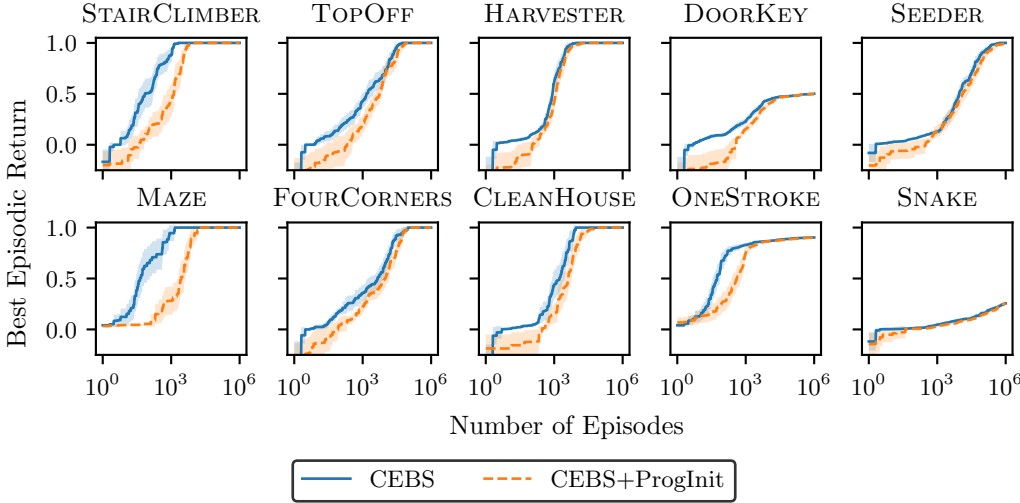

Figure 10: Episodic return performance of CEBS in LATENT SPACE with original and programmatic initialization in KAREL and KAREL-HARD problem sets. Reported mean and $95\%$ confidence interval over 32 seeds. The x-axis is represented in a log scale for better visualization.

both more robust, evidenced by the higher convergence-rate with $K = 10$, and more capable, shown by the superior convergence-rate with $K = 1,000$.

## J    CONVERGENCE RATE OF CEM AND CEBS

We further analyze the convergence rate of the LATENT SPACE by using CEM and CEBS as the search algorithm in Equation 5. Figure 12 compares the convergence rate obtained with HC (original setting), CEM, and CEBS, with $K = 64$ (neighborhood size). Although CEM and HC have a similar convergence rate across all tasks, we see that CEBS outperforms both in HARVESTER, DOORKEY, ONESTROKE and SEEDER. These results highlight the ability of CEBS to escape local optima. Despite the superior performance of CEBS, HC searching in the PROGRAMMATIC SPACE performs better than CEBS searching in the LATENT SPACE (see Figure 3).

## K    VALIDATING THE EXISTENCE OF HIGH-PERFORMING POLICIES IN LATENT SPACE

To complement the performance and topology analysis of the LATENT SPACE, we designed an additional experiment to validate the existence of high-performing policies in the space. Specifically, we are interested in confirming if policies with an episodic return larger than $0.5$ in the DOORKEY task exist in the LATENT SPACE.

We start by selecting a high-performing policy found by searching in the PROGRAMMATIC SPACE. This policy is then encoded as a latent vector to be represented in the LATENT SPACE. Note that the encoded policy does not necessarily decode to the same original policy. To account for that, we use the encoded policy as the initial candidate for an HC search on the LATENT SPACE with neighborhood size $K = 1,000$. Table 5 shows a particular choice of initial candidate that led to the discovery of a policy in the LATENT SPACE that yields an episodic return of $0.71875$ in DOORKEY.

This experiment confirms that high-performing policies exist in the LATENT SPACE. However, the convergence rate analysis shows that searching with HC in the space did not result in such policies, after considering $10,000$ different search initializations.

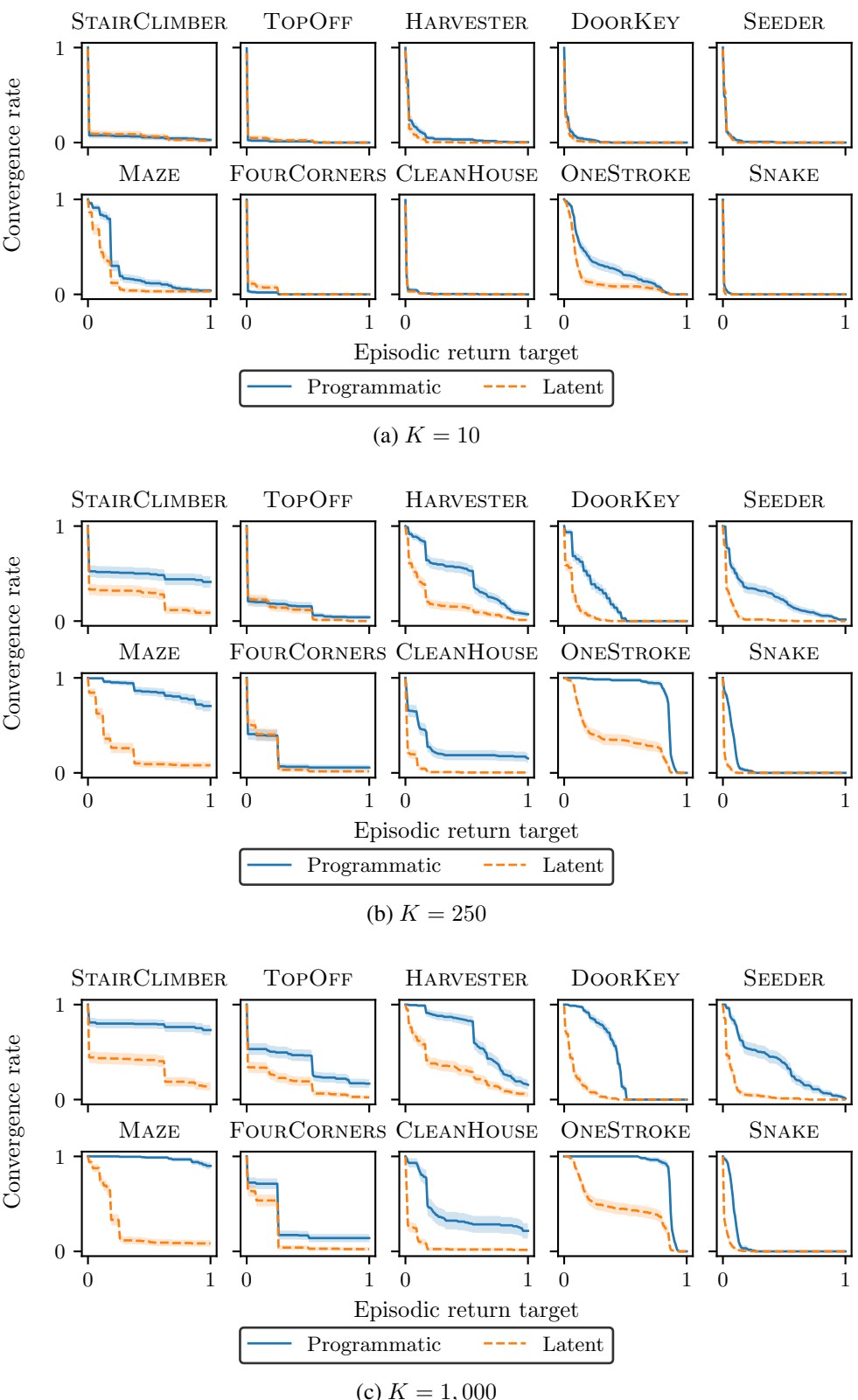

Figure 11: Convergence rate of PROGRAMMATIC SPACE and LATENT SPACE with neighborhood sizes $K = 10$ (a), $K = 250$ (b) and $K = 1,000$ (c), guided by hill-climbing. Reported mean and 95% confidence interval of estimation over a set of 250 search initializations.

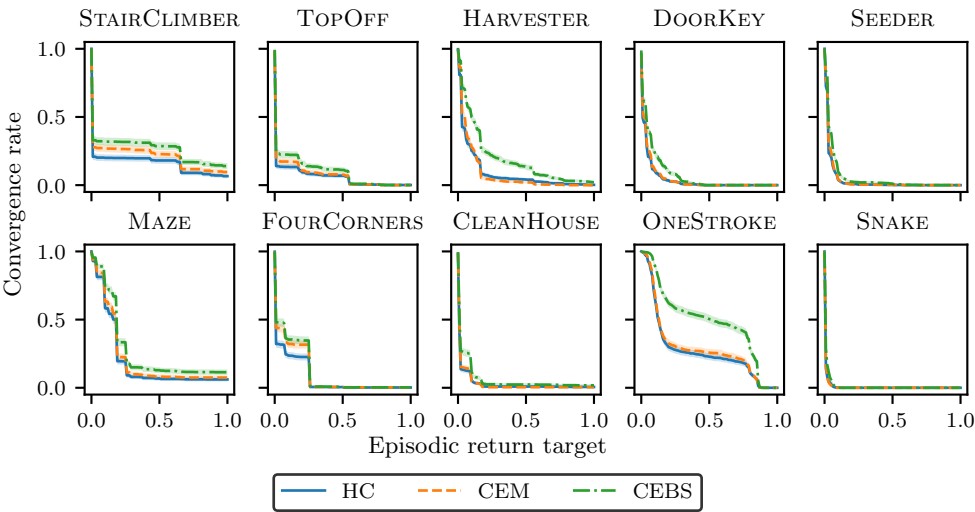

Figure 12: Convergence rate of LATENT SPACE with neighborhood size $K = 64$, guided by HC, CEM, and CEBS. Reported mean and $95\%$ confidence interval over $1,000$ seeds.

|  | Policy | DOORKEY return |
|---|---|---|
| Original initial candidate | `DEF run m( WHILE c( noMarkersPresent c) w( turnLeft pickMarker move move w) pickMarker move WHILE c( noMarkersPresent c) w( REPEAT R=17 r( move r) move turnLeft move w) putMarker m)` | 0.9375 |
| Decoded initial candidate | `DEF run m( WHILE c( noMarkersPresent c) w( turnLeft pickMarker move w) move move REPEAT R=3 r( move r) putMarker turnLeft m)` | −0.5 |
| Search result | `DEF run m( WHILE c( noMarkersPresent c) w( turnLeft move move w) pickMarker move WHILE c( noMarkersPresent c) w( turnLeft move w) WHILE c( rightIsClear c) w( putMarker w) m)` | 0.71875 |

Table 5: Initial candidate and resulting program from searching for a high-performing policy in LATENT SPACE for the DOORKEY task. The original initial candidate is defined in the PROGRAMMATIC SPACE, while the decoded initial candidate and the search result are defined in the LATENT SPACE.

