# OpenReview forum: "Reclaiming the Source of Programmatic Policies: Programmatic versus Latent Spaces"
_ICLR.cc/2024/Conference — ICLR 2024 poster_

### Official Review · Reviewer_BAZd · 2023-10-29

**Soundness:** 2 fair
**Presentation:** 2 fair
**Contribution:** 2 fair
**Rating:** 5
**Confidence:** 4

**Summary:**

This paper addresses the problem of searching programmatic policies for partially observable MDP. To this end, the paper proposes to conduct the search in the programmatic space. The experiments show the search in programmatic space will bring better numerical results with higher convergence rates. I believe this work provides an interesting point of view for program search from the program space instead of the learned latent space.

**Strengths:**

**Motivation**
The motivation for synthesizing programs as a more interpretable and generalizable representation of RL policies is convincing. This paper presents an effective method to address this problem.

**Experimental Analysis**
- Studies about the experimental result are comprehensive. This paper performs different kinds of topology-based evaluation to analyze the programmatic and latent space.
- Multiple metrics (e.g., $\rho$-similarity, behavior-similarity, identity-rate, convergence-rate) are used to evaluate the search results from the programmatic and latent space.

**Weaknesses:**

**Clarity**

- Details about the search in the programmatic space are not sufficient.
- What’s the maximum length of the sampled programs?
- What is the exact number of times the production rule s := s; s (statement chaining) can be used?
- What is the exact height of the abstract syntax tree (AST) of every program?
- The above settings should be “similar” to LEAPS based on the description of Section 3.1, but are they identical to all the settings used in LEAPS in all experiments? What are the actual numbers used for the search in the programmatic space?


**Novelty & contribution**

- Overall, I do not find enough novelty in this work but the overall effort of this paper is appreciated.


**Oversell**

- Because the local search in the programmatic space is not continuous, the initial candidate and the randomness of the programs sampled from the distribution will dominate the quality of the search results. However, there is no further detail about how the initial candidate is sampled and how many random seeds are used to evaluate the search result. Such ambiguity makes it hard to assess the robustness and efficacy of this work.
- The paper makes vague promises that are either not concrete or not trivially feasible to me. For example, the authors state in Section 5.2 that if the search algorithm fails to converge but its execution is still within the budget, an initial program is re-sampled and restart the search. I am not sure how often this kind of failure would happen. Since the budget is high (10^6) in this work, the search will be like a brute-force searching paradigm if the failure rate is high.

**Method**

- Why use fixed random seeds for the experiments as described in Section 3.1? Will the program search in the programmatic space work properly under different initial seeds? Will HC converge in all tasks in Table 1 under different random seeds?
- In the Hill Climbing search, how do the authors choose the best-seen candidate if all candidates have the same episodic return (e.g., 0.0)?
- As described in the second paragraph of Section 5.2, If an algorithm fails to converge but its execution is still within the budget, an initial program is re-sampled, and restart the search. What is the failure rate of HC and CEBS in each task in Table 1?


**Experiment details**

- How many seeds are used for each algorithm in Table 1 and Figure 3,4,5?
- The details of the initial candidate programs are missing. How is the initial candidate determined to construct the search graph for each task? How long is the initial candidate program? From what probability distribution is it sampled?

**Reproducibility**

- The implementation details are lacking from the main paper, which makes reproducing the results difficult.
- No figure or pseudocode of the Hill Climbing and the Cross Entropy Beam Search is provided for the Karel environment, which makes it hard to testify and evaluate the effectiveness of the proposed method.
- No program from the search in programmatic space is shown in the paper, making it hard to assess the efficacy of program search in the programmatic space.


**Experimental conclusions**

- The experiment about convergence analysis is not convincing and could be misleading. In Table 1, the HC can achieve a score of 0.84 on task "DoorKey" while the corresponding convergence rate is 0.0 beyond the episodic return of 0.5 in Figure 5 (A similar observation can also be found on task "Snake"). Is it a contradiction, or the result of HC in Table 1 is based on rare cases?

**Questions:**

As stated above.

---

> ### Author Response · Authors · 2023-11-14
>
> **Clarity Questions**
>
> Thank you for raising these questions. While in the original submission we had deferred the answer to these questions to the LEAPS paper, we have added an answer to all of them in the Appendix (see page 12 of the revised version).
>
> Note that all these parameters come from the LEAPS paper, so we would have a fair comparison of the search spaces.
>
> **Question:** What’s the maximum length of the sampled programs?
>
> **Answer:** 45 symbols in the program's text representation.
>
> **Question:** What is the exact number of times the production rule s := s; s (statement chaining) can be used?
>
> **Answer:** 6 times.
>
> **Question:** What is the exact height of the abstract syntax tree (AST) of every program?
>
> **Answer:** The maximum height of each program is 4.
>
> **Question:** The above settings should be “similar” to LEAPS based on the description of Section 3.1, but are they identical to all the settings used in LEAPS in all experiments?
>
> **Answer:** Yes, they are identical to all the settings used in LEAPS in all experiments. We clarify this in a sentence in Section 3.1 of the revised version and in the Appendix A.
>
> **Question:** What are the actual numbers used for the search in the programmatic space?
>
> **Answer:** Please see our answers above.
>
> **Comments on Novelty**
>
> We agree that our work does not introduce a new method and, in this regard, is not novel. Our contribution is unusual compared to most papers, as we aim to influence the research direction of the sub-field by providing a baseline missed in previous work. The results of this baseline raise fundamental questions related to how the community is using latent spaces in the context of programmatic policies. Although the contribution is unusual, we don't see it being less important than published papers proposing novel algorithms.
>
> **Comments on Oversell**
>
> **Re: Distribution of Programs.** We use the distribution of programs from the LEAPS paper to sample initial programs and to sample neighbors of a candidate program. We added a section in the Appendix describing the process used in the LEAPS work, which we adopted in our work (see page 12 of the revised version).
>
> **Re: Number of Seeds.** In our original submission we didn't use the word "seeds" to denote independent runs of the system, and that explains why the reviewer might have missed this information in the paper. In the revised version, we updated the text to always use the word "seeds". The smallest number of seeds used was 32.
>
> **Re: Brute-Force Search.** The size of the space is so large that $10^6$ is far from approaching the size of the space. The plots in Figure 5 provide the information on how often the restarts happen. Since the plots show the convergence rate $p$, the failure rate is $1 - p$. Note that restarting the search is standard in local search algorithms.

---

> ### Author Response · Authors · 2023-11-14
>
> **Comments on Method**
>
> **Re: Fixing Seed.** We only fix the seed to be able to talk about an optimization landscape. Fixing the seed is standard in any ML experiment: every time we run an experiment, we fix the seed and run the experiment; this process is repeated and results are averaged. Your comment made us realize that our wording was confusing. Since fixing seeds is standard practice in all experiments in the field, we removed that sentence from the paper.
>
> **Re: All Candidates with Same Return.** We break ties arbitrarily in all search algorithms. That is, if two candidate solutions have the same value (e.g., 0.0), then we arbitrarily select one of them.  We added a sentence in the first paragraph of Section 4 of the revised version explaining this.
>
> **Re: Failure Rate.** The failure rate can be computed from Figure 5, as it is given by $1 - p$, where $p$ is the convergence rate and provided in Figure 5. Note that Figure 5 provides $p$ for HC for both spaces and not for CEBS. This is because we are interested in the property of the space, which HC is able to provide because it terminates once it encounters a local optimum. We provide the convergence rate of CEM and CEBS in Appendix I, for completeness.
>
> **Comments on Experiment Details**
>
> **Re: Number of Seeds.** As explained above, we changed the text to explicitly talk about "seeds". See the revised version of the paper. For Table 1 we used 32 seeds, for Figures 3, 4, and 5 we used 32, 1,000, and 10,000 seeds, respectively.
>
> **Re: Information About Candidates.** We added this information in the Appendix of the revised version of the paper, where we explain the process used in the LEAPS paper that we adopted in our experiments.
>
> **Reproducibility**
>
> We will make our code available after the review process, so the results can be easily reproduced.
>
> **Re: No Pseudocode.** We added the pseudocode for Hill Climbing and CEBS to the Appendix; see pages 12-13 of the revised version.
>
> **Re: Example Programs.** We added one program for each problem domain in the Appendix; see pages 15-16 of the revised version.
>
> **Experimental Conclusions**
>
> Thank you for pointing this out. The discrepancy in the results are due to very rare events. The convergence rate experiments in Figure 5 were performed with 250 seeds, while the search budget for Table 1 and Figure 3 is 1,000,000. Rare events from the results in Table 1 and Figure 3 may not be observed with only the 250 seeds used in Figure 5. We replaced the plots in Figure 5 with plots generated with 10,000 seeds. The plots for DoorKey and Snake offer a zoomed-in highlight of the rare events. We also added a sentence in the main text explaining this discrepancy (see Section 5.3.2 of the revised version).

---

> ### Author Response · Authors · 2023-11-21
>
> We hope our responses address your questions and concerns. If the reviewer is satisfied with our answers, we would kindly ask them to consider raising the score of our paper. We would also be happy to answer any further questions you may have.

---

> > ### Comment · Reviewer_BAZd · 2023-11-22
> > **Re: Official Comment by Authors**
> >
> > I have carefully read the reviews submitted by other reviewers, and the rebuttal and the revised paper provided by the authors. I appreciate the efforts put into answering my questions and improving this submission. In that regard, I am raising my score to 5.
> >
> > Regarding the novelty, I agree with the authors that proposing this "baseline" contributes to this research community. Yet, I still believe that it would be even better if the authors could devise a method, based on the findings described in this work, that can produce capable program policies.

---

> > > ### Author Response · Authors · 2023-11-23
> > >
> > > Thank you for reading our rebuttal and updating the rating of our submission.
> > >
> > > We agree that our paper does not present a novel algorithm and we are clear about it. However, our paper still makes an important contribution, since it can change the way the community approaches the problem of learning latent spaces for programmatic policies. Future work will have to learn spaces that are better suited to search algorithms. This possibly means that the learned spaces have to be lower dimensional than the original programmatic space; if not of a lower dimension, then they might need to somehow be more conducive to search.
> > >
> > > These are very challenging questions that perhaps we can't solve by ourselves. The goal of our paper is to allow for a community effort on tackling these challenging questions. We still believe that the LEAPS line of research is very exciting and promising, but the lack of a baseline that searches in the original programmatic space is not allowing us (the community) to ask the right questions.

---

> > > > ### Comment · Reviewer_BAZd · 2023-11-23
> > > > **Re: Official Comment by Authors**
> > > >
> > > > Thanks for the comment. I am well aware of it and have increased my score because of it.

---

### Official Review · Reviewer_acX2 · 2023-10-30

**Soundness:** 2 fair
**Presentation:** 2 fair
**Contribution:** 2 fair
**Rating:** 3
**Confidence:** 3

**Summary:**

The authors introduce a simple method (direct search in programmatic space ) without any deep neural netowrks for decision problems. The mehtod outperforms the neural network baselines on Karel.

**Strengths:**

The authors introduce a simple method without any deep neural netowrks for decision problems. The mehtod outperforms the baselines on Karel.

**Weaknesses:**

1. The experiment enviornment is a rather simple one and use of knowledge of the simple grid world environment reduces strength of the proposed method.
2. Performance mismatch:  Table 1 and Figure 5 please check.
3. Potential dependance of the performance of the algorithm on initial condidtions. Please test.

**Questions:**

1. Please justify why the proposed method "is able to escape such local maxima" and "this is a property of the search space itself”. To this reviewer, it is not straightforward. Either theoretial or numerical evidences should be provided.

**Details Of Ethics Concerns:**

No problem found.

---

> ### Author Response · Authors · 2023-11-14
>
> **Comments on Summary**
>
> Contrary to what is stated, we do not introduce a new method, our paper shows that an old and simple method outperforms more complex ones. Our contribution is to show that, given the current systems, the use of latent space to search for programmatic policies is not necessary. We also provide empirical evidence of what makes the programmatic space more suitable to search than latent spaces.
>
> **Comments on Weaknesses**
>
> 1. We did not choose the problem domain nor the domain-specific language. We are comparing a simple method using the same methodology used in recent previous work. Having said that, the methodology used in previous works is fine. Karel is a challenging domain and helpful for driving research in reinforcement learning.
> 2. Could you please explain what the mismatch is? Perhaps you are referring to the convergence result providing a value of zero for high reward values for DoorKey and Snake, as reviewer BAZd pointed out. The discrepancy in the results are due to very rare events. The convergence rate experiments in Figure 5 were performed with 250 seeds, while the search budget for Table 1 and Figure 3 is 1,000,000. Rare events from the results in Table 1 and Figure 3 may not be observed with the 250 seeds used in Figure 5. We replaced the plots in Figure 5 with plots generated with 10,000 seeds. The plots for DooKey and Snake now offer a zoomed-in highlight of the rare events. We also added a sentence in the main text explaining this discrepancy (see Section 5.3.2 of the revised version).
> 3. What do you mean by "initial conditions"? Do you mean the program that is used to seed the search? We generate initial programs using the same distribution of programs used to train the latent spaces in LEAPS and HPRL. We agree that the initial program can influence the search, but testing this is beyond the goal of our study, which is to evaluate latent and programmatic spaces. Since both latent and programmatic spaces use the same distribution of programs, the comparison is fair.
>
> **Answers to Questions**
>
> **Comment:** Please justify why the proposed method "is able to escape such local maxima" and "this is a property of the search space itself”.
>
> **Answer:** The search algorithm we use in the programmatic space, Hill Climbing (HC), does not escape local optima because once it finds one, it terminates the search. The only explanation for the HC search in the programmatic space being able to find good solutions to the DoorKey problem must be a property of the search space.
>
> **Comment:** To this reviewer, it is not straightforward. Either theoretial or numerical evidences should be provided.
>
> **Answer:** We provide plenty of empirical evidence for such a claim. Figure 3 shows that HC is able to achieve average rewards higher than 0.5, the local optima the search in the latent space cannot overcome. Figure 5 of the revised version shows the convergence rate for DoorKey, showing that in a small number of runs, HC is able to obtain rewards larger than 0.5.

---

> > ### Author Response · Authors · 2023-11-21
> >
> > We hope our reply addresses your questions and concerns. If the reviewer is satisfied with our answers, we would kindly ask them to consider raising the score of our paper. We would also be happy to answer any further questions you may have.

---

> > > ### Comment · Reviewer_acX2 · 2023-11-22
> > >
> > > Thanks authors for the explanations to my questions although I am not convinced. I prefer to keep the score.

---

> > > > ### Author Response · Authors · 2023-11-23
> > > >
> > > > Thank you for responding to our rebuttal. However, it is unclear what the reviewer is not convinced of. Could you please clarify?

---

### Official Review · Reviewer_xg79 · 2023-10-31

**Soundness:** 3 good
**Presentation:** 3 good
**Contribution:** 2 fair
**Rating:** 3
**Confidence:** 4

**Summary:**

The paper compares search for programmatic policies in the policy space and the latent space. The comparison results show that the same programmatic policy space search for the same domains converges faster in the policy space than in the latent space.

**Strengths:**

It is very welcome, in my opinion, to have a paper that questions  advantages of a more complicated approach over a simple and straightforward one, and this is one such paper. The authors did an excellent job in planning, conducting, and visualizing the empirical evaluations. The paper also provides a well organized review of background and related work, making it easy to understand the problem for an outsider.

**Weaknesses:**

I think the authors miss the main point of using latent spaces instead of observable spaces for search. This is not to make the search more accurate, but rather make impossible search possible. Take, for example,  atomic physics and classical Newton's mechanics. Every problem of Newton's mechanics can, in principle, be solved within the framework of atomic physics, and the solution using atomic mechanics is likely to be more accurate, in particular when the number of atoms is relatively small. However, 1g of carbon contains ~6 * 10^23 atoms. Solving directly a problem with 10^23 variables is beyond the capacity of any modern computer; therefore, the latent space of Newton's mechanic is used instead.  There are similar examples in other areas of physics and computer science.

Same goes about learning programmatic policies. As long as one can efficiently sample the K-neighborhood of a policy in the policy space, searching in the policy space is going to be more accurate and converge better than search in latent space. However, sampling even relatively longer programs becomes increasingly difficult, and intractable for real-world problems. So, showing that latent space search is worse than observable space search if you CAN search in the observable space efficiently is a trivial result.

Practically talking, reports of the running time, of the search in total and of individual search steps, are not given in the paper, and I believe that, properly measured, that would provide proper insights. How does the running time depends on the program length? The program's branching factor? I would expect these dependencies to be quite steep. On the other hand, selecting a candidate latent vector in the latent space is fixed time.

**Questions:**

I would appreciate a detailed comparison of running times, as well as rejection rates, and everything that would provide insights on relative performance of the algorithms with the domain and policy sizes going up.

---

> ### Author Response · Authors · 2023-11-14
>
> We appreciate the reviewer's comments and insights. The reviewer has similar insights to what motivated us to write this paper, except that they misunderstand an important aspect of previous work, and we are happy to clarify the issue in this reply.
>
> In both LEAPS and HPRL, the search is performed in the latent space, but is evaluated in the programmatic space. That is, every latent vector considered in search has to be decoded into a program that is run in the environment. The reward the program obtains is used to inform the search in the latent space. The physics analogy would be valid to the LEAPS setting if the search for programmatic policies was all performed in the latent space, without the need to decode the vectors into programs, but this is not what was done in the LEAPS and HPRL work. Back to the physics analogy, the LEAPS approach would be equivalent to having to solve atomic physics problems to evaluate Newton's classic formulae. The point of our paper is to show that the effective use of latent spaces for the synthesis of programmatic policies is still an open question.
>
> The explanation above also clears your question related to sampling complexity. Sampling in the programmatic space is computationally cheaper than sampling in the latent space. The search in the latent space decodes every vector encountered in search, so the complexity is always exactly linear on the size of the program. Although the complexity is also linear for the programmatic space in the worst case, on average it will be much better. This is because the mutation-like operations performed in the programmatic space will only change some of the symbols of a program, and all other symbols are reused from the current candidate solution.
>
> We ran experiments to demonstrate this by measuring the time each search space takes to generate one neighbor from a given candidate, which was sampled from the initial distribution. Sampling from the programmatic space is more than 10x faster than sampling from the latent space, measured from generating the one-neighborhood of 1,000 initial programs. We have added these results to the paper (see Table 2 in the Appendix of the revised version).
>
> In our experiments, we gave the latent space an advantage by not showing running time results. If we had shown the results in terms of running time, the gap would be even larger.
>
> We would be happy to clarify this further if needed.

---

> ### Comment · Reviewer_xg79 · 2023-11-15
> **where the complexity comes from in search**
>
> The complexity does not come from decoding or modifying the program. It comes from exploring the search space. Complexity  of vicinity based search space exploration (all algorithms use it) is exponential with the dimensionality of the search space. The dimensionality of the search space in latent spaces is independent of the program size (can be chosen heuristically or optimized as a hyperparameter). The dimensionality of the program space is the length of the program.
>
> You (as you answered to another reviewer) compare searches on programs of 45 char max. This is smaller than the latent space representation. It is not surprising that search in a lower-dimensional space with a representation closer (identical) to the target space the search works better. It will not if the program size is 4096 char max (in the same language). Programmatic policies can be complicated, they do not always guide a toy benchmark problem.  Sometimes, they guide a surgery or an aircraft.
>
> This is a general issue with metric embeddings of non-metric spaces. I will refrain here from examples and analogies to not derail the discussion.
>
> I am not convinced that a result showing that latent space search on instances of 45 bytes in size is slower than target space search is a novel result. I was told roughly that when I took a course on AI planning, many years ago. I am telling students roughly that when I introduce the dimensionality-related challenges in search and statistical inference.

---

> > ### Author Response · Authors · 2023-11-21
> > **Uploaded a new revision**
> >
> > We have uploaded a new revision of our paper where we better explain the contribution of our work, following the messages in this thread. Please see the text highlighted in blue in the introduction.
> >
> > We hope the two paragraphs in the introduction address your questions and concerns. If the reviewer is satisfied with our answers, we would kindly ask them to consider raising the score of our paper. We would also be happy to answer any further questions you may have.

---

> ### Author Response · Authors · 2023-11-15
>
> Thank you for engaging in a discussion with us. This is very helpful!
>
> Initially we thought you were concerned with the complexity of obtaining a sample. Since the systems we evaluate require one to decode the latent vectors into programs, this complexity, in the worst case, is the same for both programmatic and latent spaces: linear on the size of the programs.
>
> However, this is not your point. Your point is related to the dimensionality and size of the spaces. We agree with you and this is one of the reasons we wrote this paper. If the latent space is larger than the programmatic space, then what are we gaining by learning latent spaces? It is unlikely that the search in the latent space will be more effective than the search in the programmatic space if both spaces are equally large. Note that we wrote "unlikely" because larger spaces could still be more conducive to search depending on the space's topology, but as we show in our paper, this isn't the case for the current published systems.
>
> What we show in our paper is that we (the scientific community) do not currently have a good way of learning latent spaces that are effective for synthesizing programmatic policies (our simple baseline easily beats them). The reconstruction loss used in current work begs for high-dimensional spaces, which makes sense, since we are asking the model to be able to reconstruct a large set of programs. How can one compress the programmatic space into a latent space while keeping only the necessary information for synthesis? Currently, the community doesn't have an answer to this question as demonstrated by our results. Moreover, we do not even know what is the "necessary information" that the latent space needs to encode.
>
> We were hoping our paper would raise the questions we are asking in this thread. While we do not answer them in the paper, we offer a baseline that will help us mark progress moving forward. Note that we are not claiming that "latent space search on instances of 45 bytes in size is slower than target space search is a novel result". Instead, we are asking: if the latent search is not more effective than our simple programmatic search, then what is the point? This baseline raises the bar and forces us (the scientific community) to ask the right questions.
>
> Given our discussions, it is clear that we can make this message more clear in our paper. However, before attempting to edit the paper, we would like to hear from the reviewer if the point of our paper makes sense to them at this point.
>
> Once again, thank you for engaging in this discussion with us.

---

### Author Response · Authors · 2023-11-14
**Summary of Revisions**

We thank all reviewers for their insightful suggestions, which have helped us substantially improve our paper.

1. **Reviewer xg79:** Reviewer xg79 offers valuable insights into the use of latent spaces, aligning with our own motivations for writing this paper. However, there is a misunderstanding about prior work; LEAPS and HPRL require decoding vectors into programs for evaluation, contrary to xg79's interpretation of a purely latent space-based search. We believe that sorting this misunderstanding is crucial to understanding our contribution.
2. **Reviewer acX2:** We found some of reviewer acX2's criticisms challenging to address due to limited information in their review. However, we responded to all points to the best of our understanding. Please let us know if we have misunderstood any aspect, and we will be happy to provide further clarification.
3. **Reviewer BAZd:** Reviewer BAZd highlighted presentation issues, notably our assumption that readers had access to the LEAPS paper. We have corrected this in the revised paper by adding the information from the LEAPS paper to our paper, as suggested by BAZd.

Please let us know if there are any other changes you'd like to see in the paper or if further clarification is needed on our end.

---

### Meta-Review · Area_Chair_RGZt · 2023-12-06

**Metareview:**

This paper attempts to highlight what seems to be a methodological error in a certain class of programmatic policy search algorithms. The central claim is that previous work on latent space embeddings does not sufficiently compare against search in the programmatic policy space directly. The experimental results show that a direct search can find programmatic policies that are more performant and lead to better convergence rates. The paper does not provide a novel search algorithm, but attempts to draw attention to the fact that this search for novelty has arguably led to an excessive focus on latent space embedding methods that are possibly less useful than the more direct approach of searching the policy space.

**Justification For Why Not Higher Score:**

The paper raises an important point about programmatic policy search techniques, but does not provide any theoretical justifications for its findings. There is also no new proposed technique presented in the paper.

**Justification For Why Not Lower Score:**

The paper clearly makes an important scientific contribution which should be published to help guide future research in the area.

---

### Decision · Program_Chairs · 2024-01-16

Accept (poster)